# A strong sense of coherence associated with reduced risk of anxiety disorder among women in disadvantaged circumstances: British population study

Olivia Remes,[1] Nicholas W J Wainwright,[2] Paul Surtees,[1] Louise Lafortune,[1] Kay-Tee Khaw,[2] Carol Brayne[1]

[1]Department of Public Health and Primary Care, Cambridge Institute of Public Health, University of Cambridge, Cambridge, UK
[2]Department of Public Health and Primary Care, University of Cambridge, Cambridge, UK

**Correspondence to**
Olivia Remes;
or260@medschl.cam.ac.uk

## ABSTRACT

**Objective** Many patients receiving medical treatment for anxiety relapse or do not improve. Research has therefore been turning to coping mechanisms as a way to decrease anxiety rates. Previously, we showed that living in a deprived area significantly increases the risk of anxiety in women, but not in men. The objective of this study is to assess whether sense of coherence (coping mechanism) buffers the influence of area deprivation on women's risk of generalised anxiety disorder using data from the European Prospective Investigation of Cancer-Norfolk.

**Design** Large, population study.

**Setting** UK population-based cohort.

**Participants** 30 445 people over the age of 40 years were recruited through general practice registers in England. Of these, 20 919 completed a structured health and lifestyle questionnaire used to assess generalised anxiety disorder and sense of coherence. Area deprivation was measured using 1991 Census data, and sense of coherence and anxiety were examined in 1996–2000. 10 183 women had data on all variables.

**Main outcome measure** Past-year generalised anxiety disorder defined according to the Diagnostic and Statistical Manual of Mental Disorders, fourth edition.

**Results** In this study, 2.6% (260/10 183) of women had generalised anxiety disorder. In those with a strong sense of coherence, area deprivation was not significantly associated with anxiety (OR 1.29, 95% CI 0.77 to 2.17). However, among women with a weak sense of coherence, those living in deprived areas were almost twice as likely to have generalised anxiety disorder compared with those living in more affluent areas (OR 1.99, 95% CI 1.37 to 2.91).

**Conclusion** The number of women living in deprived conditions is large worldwide, and significant numbers are affected by generalised anxiety disorder. Sense of coherence moderates the association between area deprivation and anxiety in women; therefore, interventions targeting coping mechanisms may need to be considered for people with anxiety.

## INTRODUCTION

Generalised anxiety disorder (GAD)[1] is one of the most common anxiety disorders in the general population.[2–4] It is characterised by

### Strengths and limitations of this study

► We used a large, population-based sample of adults over the age of 40 and adjusted for important confounders, including sociodemographics and medical history.
► We used a structured approach to measuring presence of past-year generalised anxiety disorder, and sense of coherence.
► We measured area deprivation and sense of coherence by using common, valid and theoretically sound indices.
► Because respondents were slightly more affluent and healthier than individuals living in other parts of England, our findings might not generalise to people living in extremely deprived areas.

excessive and pervasive worry about a number of areas of life, and associated symptoms, such as, restlessness, irritability, muscle tension, sleep difficulties and concentration problems.[1] If left untreated, this disorder can increase the risk for disability, impairment and suicide.[2–5] Although treatment for anxiety exists in the form of psychotherapy and pharmacotherapy, very few people who need treatment actually receive it.[6] One of the reasons for this is that physicians underdiagnose and misdiagnose those affected, and few people experiencing symptoms seek help from the clinician.[7] Low rates of help-seeking is a result of low general awareness about the disorder and treatment options, and people perceiving their anxiety to be an intractable personality trait, rather than a condition that can be treated. These problems are further compounded by the fact that even after patients are treated, many relapse, while some do not experience improvement in symptoms.[7]

While it is not known what causes anxiety, most studies on risk have focused on individual-level determinants of anxiety

disorders such as personal income, education and history of psychopathology.[8–11] However, research has shown that the environment can have a profound effect on mental health, over and above individual-level circumstances. The living context, such as, living in a deprived area, can have harmful effects for mental health independently of factors, such as sociodemographics.[12 13] Women have been reported to be particularly affected by their context or the environment in which they are living.[14 15] Women living in poor areas seem to be disproportionately affected by mental disorders.[16 17] Previously, we showed that women living in deprivation had a significantly higher risk of GAD, while this was not observed in men.[16] If women are living in an area with low socioeconomic circumstances, they are more likely to be exposed to the stress and strain that arises from deprivation.[14] Exposure to stress can then increase the risk for inflammation and hypothalamic-pituitary-adrenal axis dysregulation, which may lead to the development of GAD.[18 19]

To reduce the risk of mental disorders among women exposed to disadvantage or adversity, coping skills need to be considered. In particular, sense of coherence (SOC), which is a way of viewing life as predictable, manageable and meaningful, can lower the risk for poor health outcomes.[20–22] Also, SOC is a flexible and adaptive dispositional orientation which enables coping with stressful situations.[20 21]

Two systematic reviews[22 23] showed that SOC is linked to quality of life and perceived health. A strong SOC is related to good physical and self-perceived health, and is negatively associated with anxiety, depression and post-traumatic stress disorder.[23] In the European-Prospective Investigation of Cancer-Norfolk (EPIC-Norfolk) study of over 18 000 people, a strong SOC was linked to a 20% lower risk of all-cause mortality in adults.[24] SOC has also been shown to moderate the influence of disadvantage and adversity on mental health outcomes. In a study of people who had faced early childhood deprivation during the Holocaust, SOC moderated the association between early life deprivation and post-traumatic stress in old age.[25] A strong SOC can therefore be an important coping resource for remaining healthy.

Previously,[16] we have shown that women living in deprived areas were at increased risk for GAD. The stress of living in deprivation was harmful for women's mental health, while this association with deprivation was not apparent in men. For this reason, this study will focus on women. The objective of this study is to determine whether SOC moderates the link between area deprivation and GAD in women using a large, longitudinal, population cohort.

## METHODS
### Study population
Data were drawn from the population-based EPIC-Norfolk study, whose methods have been described by previous research.[26] Between 1993 and 1997, 30 445 participants aged 40–74 years living in Norwich and the surrounding towns and rural areas were identified through general practice age-sex registers (77 630 people were initially approached to join EPIC-Norfolk). In 1993–1997, 30 445 participants consented to join the study and filled out a postal Health and Lifestyle (HLQ) questionnaire. The HLQ contained questions on sociodemographics, including gender, marital status, highest educational attainment, employment, as well as self-reported physician diagnoses of physical diseases. To derive a measure of area deprivation, participants' postal codes were linked to the 1991 Census.[27] During the 1993-2000 time period, respondents completed self-reported postal questionnaires if they were still alive, remained on the study's mailing list, and had a valid mailing address.

All participants recruited in 1993-1997 and who completed the HLQ were eligible to be included in our study; those who completed a psychosocial questionnaire during follow-up were eligible for inclusion in our analysis.

### Assessment of GAD: outcome
In 1996–2000, 20 919 men and women completed a Health and Life Experiences Questionnaire (HLEQ)[28] used to identify those meeting criteria for Diagnostic and Statistical Manual of Mental Disorders, fourth version (DSM-IV) GAD. The outcome in this research was past-year GAD. The HLEQ captured the onset and offset timings of episodes of GAD.[29] Past-year GAD was present if participants reported at least one episode that had offset within 12 months of administration of the HLEQ. DSM-IV GAD was defined as uncontrollable, excessive worry for 6 months or longer on most days than not that resulted in life interference and help-seeking. In addition, at least three ancillary symptoms needed to have been present: restlessness, irritability, muscle tension, fatigue, trouble concentrating because of worry, mind going blank, trouble falling asleep, trouble staying asleep and feeling keyed up or on edge. Of those who completed the HLEQ, 461 met criteria for past-year DSM-IV GAD.

### Assessment of potential confounders
Covariates were chosen a priori based on previous literature (their links to anxiety[7 30–33] and deprivation[16 34]). The baseline HLQ was used to ascertain gender, education (highest level of education attained: no qualifications, educated to age 16 years, educated to age 18 years or educated to degree level), marital status (single, married, widowed, separated, divorced), employment (yes, no) and self-reported physician diagnoses of major medical conditions (self-reported asthma, bronchitis, allergies, hay fever, stroke, heart attack, cancer, diabetes, thyroid conditions, arthritis). Social class (professionals, managerial and technical occupations, skilled workers divided into non-manual and manual, partly skilled workers and unskilled manual workers) was determined using the Computer-Assisted Standard Occupational Coding.[35]

The HLEQ was used to derive participant age, determine presence of lifetime major depressive disorder (MDD) according to the DSM-IV and disability measures based on the Medical Outcomes Study Short Form 36 (SF-36). To identify level of disability, the physical component summary score (PCS) of the SF-36, a widely used, validated self-assessment instrument was used. Higher scores represent better health.[36] PCS scores were dichotomised above and below the median.

### Assessment of area deprivation: exposure

To examine area deprivation, we used the Townsend Index.[37 38] This index consists of the following four variables derived from the 1991 Census and obtained at the level of the enumeration district: (1) percentage of economically active residents over age 16 who are unemployed, (2) percentage of households that do not possess a car, (3) percentage of private households that are not owner occupied and (4) percentage of private households that are overcrowded (have more than one person per room). For each variable, Z-scores were obtained by dividing the mean by the SD (across enumeration districts in England). The Z-values of the variables were summed to produce a Townsend Index score, with positive values of the index indicating areas that are more deprived and negative values representing areas that are less deprived; 0 corresponds to the national mean. Record linkage between participant postal codes and enumeration districts was conducted; respondents were considered to live in deprived areas depending on the Townsend Index score that their enumeration district received.

### Ascertainment of SOC

The HLEQ included an SOC questionnaire[39] that enquired about three items assessing each of the SOC constructs. The following questions were used to assess each construct:

#### Comprehensibility

Do you usually feel that the things that happen to you in your daily life are hard to understand?

#### Manageability

Do you usually see a solution to problems and difficulties that other people find hopeless?

#### Meaningfulness

Do you usually feel that your daily life is a source of personal satisfaction?

Participants were given the choice of responding to these questions with yes, usually; yes, sometimes and no. Comprehensibility was reverse scored, and all items were then added together to produce a total SOC scale ranging from 0 to 6. Higher scores represent weaker SOC.

### Statistical analysis

Characteristics of the participants were compared by GAD status. We used correlated data analysis to assess the association between individual- and area-level measures and GAD in women with low and high SOC. A population-average model was built, and it took into account the potential correlation arising from the clustering of individuals within enumeration districts. To estimate the population-average effect of the risk factors of interest on past-year GAD, generalised estimating equations (GEE) were used. As past-year GAD is a dichotomous outcome (yes/no) and the intracluster correlation is believed to be equal, GEE with a logit link and an exchangeable correlation structure was the method of choice. Adjusted ORs and 95% CIs were derived. Multivariate logistic regression was also undertaken and compared with the findings based on correlated data analysis.

Individual-level measures consisted of sociodemographic and health variables, whereas the area-level measure was the Townsend Index. The Townsend Index variable was dichotomised using a cut-point of 0 (representing the national average).

SOC was split at the median (of 2) and participants below this cut-point were classified as strong on SOC, while those above this cut-point had a weak SOC. The interaction between area deprivation and SOC in women was assessed. After this, analyses were conducted separately for those with strong and weak levels of SOC. First, unadjusted effect estimates were ascertained. Subsequently, models were built that adjusted for (1) age, educational attainment, marital status, social class and employment; then for (2) age, educational attainment, marital status, social class, employment and lifetime MDD and lastly for (3) age, educational attainment, marital status, social class, employment, lifetime MDD, physical diseases and disability level. Age was assessed as a categorical variable. A complete case analysis was conducted. The brackets show the reference categories that were used for each categorical variable entered in the models—age: young (<65 years) versus old (≥65 years) [ref]; education: high [ref] versus low; marital status: married [ref] versus not married; social class: non-manual [ref] versus manual; employed: no versus yes [ref]; lifetime MDD: no [ref] versus yes; deprivation: no [ref] versus yes; prevalent physical disease: no [ref] versus yes; disability level: low [ref] versus high. The literature guided the selection of reference categories; choosing other groupings for the potential confounders would not have changed the findings. It was not possible to re-classify the GAD variable, and area deprivation was analysed according to the literature.

This is how we arrived at the study size: of the 30 445 who completed the baseline HLQ, we kept respondents (both men and women) who completed the HLEQ (20 919), and of these, we kept only women with complete data on all covariates (10 183) (figure 1).

### Patient involvement

There were no patients involved in the development of the research question and outcome measures, the design of the study or the recruitment to and conduct of the study.

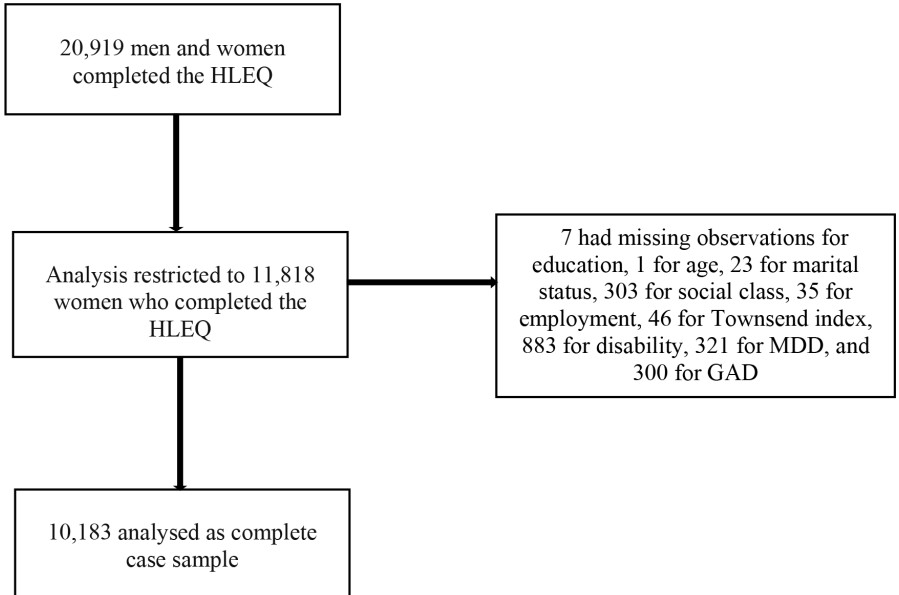

**Figure 1** Flow chart of European Prospective Investigation of Cancer (EPIC)-Norfolk cohort. This is a flow chart showing the number of participants at each study stage: the total number who completed the psychosocial Health and Life Experiences Questionnaire (HLEQ) in the EPIC-Norfolk study, the number of women who filled out the HLEQ and with data on all variables. The EPIC-Norfolk study consists of middle-aged and older British people. GAD, generalised anxiety disorder.

## RESULTS

Of the 77 630 people identified through general practices in Norfolk and invited to participate in the research, 30 445 consented.[26] The characteristics of participants versus non-participants are compared in online supplementary appendix 1; compared to those who did not take part, responders comprised slightly more women and were slightly younger. 20 919 out of the 30 445 people who consented at baseline completed the HLEQ during follow-up.[26 29] Of those who completed the HLEQ, 10 183 women had complete data on all variables and were thus included in the analysis. The number of missing observations for each variable was: 1 for age, 7 for education, 23 for marital status, 303 for social class, 35 for employment, 46 for Townsend Index, 883 for disability, 321 for MDD and 300 for GAD. Participants were assessed between 1993 and 2000 (followed for 7 years) (figure 1).

In 1996–2000, GAD was present in 260 out of 10 183 (2.6%) women. Table 1 shows sociodemographic and health status characteristics for women with a weak and strong SOC.

Among women with a weak SOC, those who also had GAD were more likely to be younger, have high educational attainment, non-manual social class, live in areas of high deprivation and have had pre-existing health conditions or show prevalent physical disease, high disability and lifetime MDD. In the group with strong SOC, similar patterns were found as for those with anxiety.

During the 7-year follow-up period, there were a total of 260 GAD cases in women. A weak SOC was found in 2991 women, while a strong SOC was present in 7192 women. When the interaction between area deprivation and SOC was assessed, the P value was 0.226. When area

deprivation was regressed against SOC in a fully adjusted model, the P value was 0.368; and when area deprivation and SOC were introduced in a fully adjusted model with GAD as the outcome, the P values for these explanatory variables were 0.0005 and <0.0001, respectively.

Tables 2 and 3 show the unadjusted and adjusted ORs (models A–C) associated with GAD in those with a weak and strong SOC, respectively.

Analyses that adjusted for age, education, marital status, social class and employment status showed that area deprivation was significantly associated with increased risk for GAD in women with a weak SOC (OR 2.05, 95% CI 1.43 to 2.94) (table 2), but area deprivation was not significantly associated with anxiety in those with strong SOC (OR 1.43, 95% CI 0.86 to 2.39) (table 3). In women with a weak SOC (table 2), further adjustment for lifetime MDD slightly attenuated the effect estimate, although the association between area deprivation and anxiety remained highly significant (OR 1.97, 95% CI 1.35 to 2.88). When prevalent physical disease and disability level were added to the final model, the effect estimate remained almost unchanged compared with the previous model; among women with poor coping skills, those living in deprived areas had a 99% higher likelihood of having anxiety than women living in less deprived areas (OR 1.99, 95% CI 1.37 to 2.91). For women with a strong SOC (table 3), area deprivation was associated with a small increased risk of having GAD in progressively adjusted models; however, none of the effect estimates reached statistical significance. In the fully adjusted model, women with a strong SOC and living in deprivation had a 29% higher chance of having GAD compared with women living in less

**Table 1**  Distribution of characteristics for women (n=10 183) with weak and strong SOC who completed the HLEQ questionnaire in the European Prospective Investigation of Cancer (EPIC)-Norfolk cohort

| Characteristic | Weak SOC | | Strong SOC | |
|---|---|---|---|---|
| | Number with characteristic | Percentage and number with past-year GAD | Number with characteristic | Percentage and number with past-year GAD |
| Sociodemographics | | | | |
| Age (years) | | | | |
| <65 | 1995 | 6.8 (136) | 4732 | 1.6 (78) |
| ≥65 | 996 | 2.7 (27) | 2460 | 0.8 (19) |
| Education* | | | | |
| Low | 1358 | 4.6 (62) | 2619 | 0.8 (21) |
| High | 1633 | 6.2 (101) | 4573 | 1.7 (76) |
| Marital status | | | | |
| Married | 2060 | 5.5 (113) | 5590 | 1.2 (69) |
| Not married† | 931 | 5.4 (50) | 1602 | 1.7 (28) |
| Social class‡ | | | | |
| Manual | 1261 | 4.9 (62) | 2508 | 1.1 (27) |
| Non-manual | 1730 | 5.8 (101) | 4684 | 1.5 (70) |
| Employed | | | | |
| Yes | 1178 | 5.6 (66) | 2852 | 1.4 (40) |
| No | 1813 | 5.4 (97) | 4340 | 1.3 (57) |
| Townsend Index | | | | |
| Deprivation | | | | |
| Yes (>0) | 534 | 8.4 (45) | 1083 | 1.8 (19) |
| No (≤0) | 2457 | 4.8 (118) | 6109 | 1.3 (78) |
| Health status | | | | |
| Prevalent physical disease | | | | |
| Yes§ | 1683 | 6.1 (103) | 3922 | 1.8 (70) |
| No | 1308 | 4.6 (60) | 3270 | 0.8 (27) |
| Disability level | | | | |
| High¶ | 1717 | 6.2 (107) | 3493 | 1.8 (64) |
| Low | 1274 | 4.4 (56) | 3699 | 0.9 (33) |
| Lifetime MDD | | | | |
| Yes | 737 | 13.8 (102) | 1180 | 5.4 (64) |
| No | 2254 | 2.7 (61) | 6012 | 0.5 (33) |

*High education: O-level, A-level, degree; low education: refers to no education.

†Single, divorced, separated, widowed.

‡Manual: skilled manual, semi-skilled, non-skilled; non-manual: professionals, managerial, skilled non-manual.

§Prevalent physical disease: respiratory disease (asthma and bronchitis), allergies (allergies and hay fever), stroke, heart attack, cancer, diabetes, thyroid conditions, arthritis.

¶Below the median PCS value of 50.6.

GAD, generalised anxiety disorder; HLEQ, Health and Life Experiences Questionnaire; MDD, major depressive disorder; PCS, physical component summary score; SOC, sense of coherence.

deprived areas, but this did not reach statistical significance (OR 1.29, 95% CI 0.77 to 2.17).

Similar results emerged when logistic regression was used in these models instead of GEE, which indicates that the intraclass correlation has a negligible effect (strong SOC: OR 1.29 (95% CI 0.77 to 2.18) and weak SOC: OR 1.99 (95% CI 1.36 to 2.93)).

We carried out multiple imputations for missing data (see online supplementary appendix 2); the effect estimate became slightly stronger for women with a weak SOC and living in deprivation (OR 2.28, 95% CI 1.62 to 3.21), and the association between deprivation and anxiety become even weaker for women with a strong SOC (OR 1.14, 95% CI 0.69 to 1.88).

**Table 2** OR for women with a weak SOC who completed the HLEQ questionnaire in 1996–2000 (women with weak SOC sample size=2991)

| Characteristic | OR and 95% CI | | | | P value for model C |
| --- | --- | --- | --- | --- | --- |
| | Unadjusted | Model A* | Model B† | Model C‡ | |
| Sociodemographics | | | | | |
| Age (years) | | | | | |
| <65 | 2.63 (1.72 to 4.00) | 3.35 (2.10 to 5.34) | 2.49 (1.54 to 4.02) | 2.67 (1.65 to 4.32) | <0.0001 |
| ≥65 | 1.00 | 1.00 | 1.00 | 1.00 | |
| Education§ | | | | | |
| Low | 0.73 (0.52 to 1.00) | 0.77 (0.55 to 1.08) | 0.83 (0.58 to 1.17) | 0.83 (0.58 to 1.17) | 0.287 |
| High | 1.00 | 1.00 | 1.00 | 1.00 | |
| Marital status | | | | | |
| Married | 1.00 | 1.00 | 1.00 | 1.00 | |
| Not married¶ | 0.98 (0.69 to 1.38) | 1.06 (0.74 to 1.51) | 0.86 (0.59 to 1.25) | 0.85 (0.59 to 1.23) | 0.392 |
| Social class** | | | | | |
| Manual | 0.83 (0.60 to 1.15) | 0.81 (0.58 to 1.15) | 0.83 (0.59 to 1.18) | 0.81 (0.57 to 1.15) | 0.231 |
| Non-manual | 1.00 | 1.00 | 1.00 | 1.00 | |
| Employed | | | | | |
| Yes | 1.00 | 1.00 | 1.00 | 1.00 | |
| No | 0.95 (0.69 to 1.31) | 1.55 (1.09 to 2.19) | 1.42 (0.99 to 2.04) | 1.33 (0.92 to 1.91) | 0.126 |
| Townsend Index | | | | | |
| Deprivation | | | | | |
| Yes (>0) | 1.82 (1.28 to 2.61) | 2.05 (1.43 to 2.94) | 1.97 (1.35 to 2.88) | 1.99 (1.37 to 2.91) | 0.0004 |
| No (≤0) | 1.00 | 1.00 | 1.00 | 1.00 | |
| Health status | | | | | |
| Lifetime MDD | | | | | |
| Yes | 5.77 (4.15 to 8.03) | | 5.20 (3.68 to 7.34) | 5.06 (3.58 to 7.15) | <0.0001 |
| No | 1.00 | | 1.00 | 1.00 | |
| Prevalent physical disease†† | | | | | |
| Yes | 1.36 (0.98 to 1.88) | | | 1.20 (0.84 to 1.70) | 0.316 |
| No | 1.00 | | | 1.00 | |
| Disability level | | | | | |
| High‡‡ | 1.45 (1.04 to 2.01) | | | 1.50 (1.04 to 2.15) | 0.030 |
| Low | 1.00 | | | 1.00 | |

*Adjusted for sociodemographics (age, education, marital status, social class, employment).
†Adjusted for sociodemographics, lifetime MDD.
‡Adjusted for sociodemographics, lifetime MDD, prevalent physical disease and disability.
§High education: O-level, A-level, degree; low education: refers to no education.
¶Not married: single, divorced, separated, widowed.
**Manual: skilled manual, semi-skilled, non-skilled; non-manual: professionals, managerial, skilled non-manual.
††Prevalent physical disease: respiratory disease (asthma, bronchitis), allergies (allergies, hay fever), stroke, heart attack, cancer, diabetes, thyroid conditions, arthritis.
‡‡Below the median PCS value of 50.6.
HLEQ, Health and Life Experiences Questionnaire; MDD, major depressive disorder; PCS, physical component summary score; SOC, sense of coherence.

## DISCUSSION

In this large, population-based study, we found that area deprivation significantly increased the risk for GAD in women, but particularly in those with poor coping skills. Coping skills or SOC appeared to moderate the association between area deprivation and anxiety. SOC was based on a 3-item scale, with modest internal reliability (Chronbach's α=0.35)[20] and this variable was dichotomised. Although it may be useful to additionally employ a continuous SOC measure, we dichotomised this variable because previous literature had done so as well.[24]

Women living in deprivation and with poor coping or a weak SOC were at a particularly high risk for having anxiety after controlling for important confounders. Although

**Table 3** OR for women with a strong SOC who completed the HLEQ questionnaire in 1996–2000 (women with a strong SOC sample size=7192)

| Characteristic | OR and 95% CI | | | | P value for model C |
| --- | --- | --- | --- | --- | --- |
| | Unadjusted | Model A* | Model B† | Model C‡ | |
| **Sociodemographics** | | | | | |
| Age | | | | | |
| <65 | 2.15 (1.30 to 3.56) | 2.58 (1.48 to 4.50) | 1.89 (1.06 to 3.38) | 2.13 (1.18 to 3.85) | 0.0118 |
| ≥65 | 1.00 | 1.00 | 1.00 | 1.00 | |
| Education§ | | | | | |
| Low | 0.48 (0.29 to 0.78) | 0.54 (0.33 to 0.89) | 0.59 (0.36 to 0.98) | 0.59 (0.35 to 1.00) | 0.0483 |
| High | 1.00 | 1.00 | 1.00 | 1.00 | |
| Marital status | | | | | |
| Married | 1.00 | 1.00 | 1.00 | 1.00 | |
| Not married‡‡ | 1.42 (0.91 to 2.22) | 1.56 (0.99 to 2.47) | 1.25 (0.78 to 2.01) | 1.22 (0.76 to 1.96) | 0.4131 |
| Social class¶ | | | | | |
| Manual | 0.72 (0.46 to 1.12) | 0.84 (0.53 to 1.34) | 0.86 (0.53 to 1.39) | 0.83 (0.52 to 1.35) | 0.4592 |
| Non-manual | 1.00 | 1.00 | 1.00 | 1.00 | |
| Employed | | | | | |
| Yes | 1.00 | 1.00 | 1.00 | 1.00 | |
| No | 0.94 (0.62 to 1.41) | 1.46 (0.94 to 2.26) | 1.44 (0.92 to 2.25) | 1.25 (0.79 to 1.97) | 0.3461 |
| **Townsend Index** | | | | | |
| Deprivation | | | | | |
| Yes (>0) | 1.38 (0.83 to 2.29) | 1.43 (0.86 to 2.39) | 1.32 (0.79 to 2.21) | 1.29 (0.77 to 2.17) | 0.3366 |
| No (≤0) | 1.00 | 1.00 | 1.00 | 1.00 | |
| **Health status** | | | | | |
| Life-time MDD | | | | | |
| Yes | 10.39 (6.79 to 15.89) | | 9.32 (6.05 to 14.35) | 8.58 (5.53 to 13.31) | <0.0001 |
| No | 1.00 | | 1.00 | 1.00 | |
| Prevalent physical disease** | | | | | |
| Yes | 2.18 (1.40 to 3.41) | | | 1.72 (1.10 to 2.71) | 0.0185 |
| No | 1.00 | | | 1.00 | |
| Disability level | | | | | |
| High†† | 2.07 (1.36 to 3.16) | | | 1.92 (1.21 to 3.05) | 0.0059 |
| Low | 1.00 | | | 1.00 | |

*Adjusted for sociodemographics (age, education, marital status, social class, employment).
†Adjusted for sociodemographics lifetime MDD.
‡Adjusted for sociodemographics lifetime MDD, prevalent physical disease and disability.
§High education: O-level, A-level, degree; low education: refers to no education.
¶Manual: skilled manual, semi-skilled, non-skilled; non-manual: professionals, managerial, skilled non-manual.
**Prevalent physical disease: respiratory disease (asthma, bronchitis), allergies (allergies, hay fever), stroke, heart attack, cancer, diabetes, thyroid conditions, arthritis.
††Below the median PCS value of 50.6.
‡‡Not married: single, divorced, separated, widowed.
HLEQ, Health and Life Experiences Questionnaire; MDD, major depressive disorder; PCS, physical component summary score; SOC, sense of coherence.

women with a strong SOC showed a slight increased risk of anxiety if living in disadvantaged circumstances, the association between area deprivation and GAD was statistically non-significant in women who were able to cope well and the effect estimate was much smaller than that of the former group (women with poor coping). A statistically significant association between area deprivation and GAD persisted in women with a weak SOC after adjustment for age, marital status, education level, social class, employment status, MDD, chronic physical diseases and disability. In contrast, having a strong SOC seemed to be protective for women living in deprived areas. Having a strong SOC rendered the association between area deprivation and anxiety statistically non-significant.

Although the interaction between area deprivation and SOC was not statistically significant, the effect estimates do suggest that there are differences between women with low and high SOC—nevertheless, these differences are rather small. Our study sheds light on the potential importance of SOC when it comes to mitigating the risks of anxiety. Future research should replicate our study with a larger number of anxiety cases, perhaps by measuring 'total' or 'any' anxiety rather than individual disorders, such as GAD.

Deprived areas are often associated with low social integration and poor social control. Emile Durkheim showed that low social integration can lead to a sense of meaninglessness among individuals, and this can give rise to poor mental health and suicide.[40] SOC is a way of viewing life as meaningful and comprehensible, and our study shows that SOC can moderate the association between area deprivation and GAD in women.

### Strengths and limitations of this study, and future research

This is the largest, population-based study of the association between area deprivation and GAD in women, and to determine whether coping resources or SOC moderates the association between area deprivation and anxiety. We had access to a large sample of over 10 000 women living in the community. We used a measure of anxiety defined according to the DSM-IV. Although GAD affects a substantial number of people, even more experience subthreshold cases of anxiety disorders. Subthreshold cases have also been associated with impairment and disability; therefore, future research should assess associations with subclinical anxiety.

We used detailed health and lifestyle questionnaires to extract information on demographics, social class and major chronic physical diseases, and controlled for these factors in our analyses. We used a validated and reliable measure of disability, which we adjusted for in our models. We used self-reported physician diagnoses to ascertain history of chronic physical diseases, though this might give rise to three issues. First, residual confounding may be present: diseases associated with deprivation and anxiety might not have been captured. Second, medical diagnoses were not verified by clinicians, leading to possible misclassification. Third, chronic physical diseases may have been under-reported, leading to misclassification bias and attenuation of effect estimates. We may have overadjusted our models with the inclusion of disability, because this might be part of the expression of psychiatric illness. This may have reduced effect estimates. Our objective was to assess the links between deprivation, SOC and anxiety in women. Although it was out of scope for the present study, we were unable to examine the same objectives in men: there were very few men with a strong SOC living in deprivation and with GAD. Therefore, analyses in this subgroup would not have been robust. Future studies should undertake this assessment. It should also be mentioned that the internal consistency of the 3-item SOC scale, as measured by Chronbach's α,

was 0.35.[20] The number of items forming this scale was small, and this may have partially contributed to a low internal consistency. Also, the researchers who developed the scale noted that the reliability and validity of this instrument are satisfactory.[20 41] Despite this, it was a limitation that we did not have a longer measure with higher reliability, such as the SOC-13 or SOC-29.[41] The SOC was dichotomised—it may be useful to additionally use a continuous measure of SOC, but we dichotomised it because research on coping had done so as well.[24]

At baseline, people who consented to take part in EPIC-Norfolk agreed to fill out detailed health and lifestyle questionnaires over the duration of the study period; therefore, healthy volunteer effect may have biased our findings. Participants in EPIC-Norfolk tend to be somewhat healthier and more affluent than the general population, therefore, results from this study cannot be generalised to extremely deprived areas. If the most deprived areas would have been included, we would expect the association between area deprivation and anxiety to be even stronger in women with a weak SOC. Also, when comparing the age and gender distributions of participants versus non-participants (see online supplementary appendix 1), we found that there were slightly more women and slightly younger people who chose to take part in the study.

Also, it may be that participants with poorer mental health may have moved to more deprived neighbourhoods; however, reverse causality seems unlikely as an explanation for our findings. In addition, deprivation was measured before anxiety in this study; however, SOC was examined at the same time point as GAD, rendering this study cross-sectional.

Although this cross-sectional, observational research cannot confirm that living in a deprived area causes GAD in women with a weak SOC, the analysis is rigorous and is a reasonable method of examining the relationship between these variables. When we conducted multiple imputations, the effect estimate for women with a weak SOC became even greater, and among women with a strong SOC, it attenuated towards the null. Our study provides a valuable step forward and is the first to shed light on the importance of coping in people with GAD living in disadvantaged circumstances.

### Comparison with other studies

This is the largest, population-based study to consider the association between area deprivation and GAD in women, and to determine whether SOC moderates this association. Most of the literature on coping and SOC is limited. A number of studies have small sample sizes, and measure people's coping abilities in relation to feelings of stress, history of stressful life events or exposure to stressful circumstances, such as wars. There is a paucity of research examining the living context, such as area deprivation, and no studies have assessed whether the link between area-level circumstances and anxiety disorders can be moderated by coping mechanisms. Much of the

literature on coping uses highly select samples; therefore, results cannot be generalised to the larger population. Also, incomplete adjustment of covariates makes it difficult to determine whether findings from studies are not better explained by the residual effect of other factors that have not been accounted for, such as personal socioeconomic circumstances. Across studies, there is large heterogeneity in the definitions used to define coping, with many focusing on factors, such as hardiness, optimism and negative emotions, rather than SOC. In sum, it is difficult to understand the links between the living context, coping abilities and mental health from the literature; however, the studies that have been conducted are a good starting point.

A UK study of over 3000 people[42] showed that SOC was linked to self-rated health. Research on people living in Negev communities in Israel showed that those exposed to severe stress-provoking situations, but who had coping resources, were at low risk for stress: the higher the SOC, the lower the chance of reporting stress.[43] In a study of French adults,[44] SOC buffered the effect of adversity on psychological well-being. In another study of Holocaust survivors,[25] SOC moderated the association between early childhood deprivation and post-traumatic stress in old age. Both of these latter studies, however, were small and failed to adjust for important confounders, such as sociodemographic factors and disability; also, the French study did not examine individual psychiatric disorders. In the study on child Holocaust survivors,[25] exposure to trauma was measured in early life, while post-traumatic stress in old age. Since participants were required to report traumas experienced in childhood, this might have led to recall bias. Our study expands on previous research and is the first to investigate the moderating effect of coping skills (SOC) on the risk of developing GAD in women living in deprived circumstances.

### Mechanism of effect

Living in a deprived area can increase anxiety in women because of biological and social factors.[16] The stress of living in deprivation can increase the risk for inflammation and HPA-axis dysregulation, which can lead to GAD.[18 19] This, combined with the multiple roles that women are increasingly taking on (income earner, childbearer and carer of elderly relatives), means that coping is particularly relevant for women living in disadvantaged circumstances. A strong SOC is linked to high quality of life, and good physical and mental health.[22 23] Our study shows that SOC can buffer the effect of area deprivation on risk of anxiety.

### Implications

The number of people living in deprived conditions is large worldwide, and significant numbers will have been affected by GAD.[45] For the first time, we show that SOC moderates the association between area deprivation and anxiety in women. Future research should replicate our analysis using larger samples and determine the specific components of SOC that attenuate the effect of deprivation on mental health. Interventions can then be developed to target components of SOC to strengthen people's coping resources. Treatment for GAD exists, with psychotherapy and pharmacotherapy being commonly prescribed. However, success rates are fairly low, patients relapse and some fail to experience any symptom improvement. Costs to the healthcare system related to anxiety are substantial. Therefore, targeting people's coping resources could represent another option for people with anxiety, including those who do not experience symptom improvement following commonly prescribed therapies. Targeting SOC could also be worthwhile for people who have faced extreme circumstances and adversity, and who may have difficultly exploring past traumas and reliving memories during psychotherapy. Interventions should take these findings into account, and mental health policy should also consider improving living environments to decrease the burden of anxiety in women.

**Contributors** OR (corresponding author) had the idea for and conducted the analysis, and wrote the article. CB critically reviewed drafts of the manuscript and provided input into the analysis, K-TK edited versions of the paper; PS and NW provided feedback into the analysis and reviewed the final draft of the paper. OR, CB, K-TK, LL, PS and NWJW contributed to the interpretation of data for the work, agreed to be accountable for all aspects of the work, gave final approval of the version to be published and made substantial contributions to the analysis and interpretation of data. OR, CB, K-TK, LL, PS and NW have seen and approved the final version. OR, CB, K-TK, LL, PS and NW had full access to all the data in the study and take responsibility for the integrity of the data and the accuracy of the data analysis. OR acts as guarantor of the study.

**Funding** This work was supported by the Medical Research Council UK (grant number SP2024-0201 and SP2024-0204) and Cancer Research UK (grant number G9502233).

**Competing interests** OR received a PhD studentship from the National Institute for Health Research.

**Patient consent** Obtained.

**Ethics approval** The study has ethics committee approval from Norfolk Ethics Committee (Rec Ref: 98CN01).

**Provenance and peer review** Not commissioned; externally peer reviewed.

**Data sharing statement** No additional data available.

**Author note** Nick Wainwright is a retired author.

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
