## [Reviewer comments · BMJ Open]

ARTICLE DETAILS

TITLE (PROVISIONAL)	Sense of coherence as a coping mechanism for women with anxiety living in deprivation: British population study
AUTHORS	Remes, Olivia; Wainwright, Nicholas; Surtees, Paul; LaFortune, Louise; Khaw, Kay-Tee; Brayne, Carol

VERSION 1 – REVIEW

REVIEWER	Christophe Leys Université Libre de Bruxelles - Belgium
REVIEW RETURNED	20-Sep-2017

GENERAL COMMENTS	Authors conducted a study on a cohort of 20921 participants that completed a questionnaire assessing generalised anxiety disorder (GAD), sense of coherence (SOC), deprivation and various potential covariates. They describe a moderation model showing that, in a deprivation environment, women with a low sense of coherence suffer from generalised anxiety disorders more often than women with a high sense of coherence (even when the effect of some potential confounding variables was neutralized). Although the study bears on a very large sample size, many serious shortcomings jeopardize authors results. The main concerns being a (very) poor measure of SOC and a lack of innovation (the link between adversity and GAD is well known and the role of SOC has often been identified, although sometimes as mediator and sometimes as moderator). For these reasons I suggested rejection, the rest could be overcome by a major revision. Please find below some comments : - Please provide a short description of general anxiety disorders (definition, symptoms) and some statistics about its prevalence.- Please explain why deprivation is relevant to study? Is it a trauma? Is it a significant source of stress? Is it something else? Why would it yield GAD? Why not, I don't know, depression, addiction, eating disorders or PTSD (that has been removed from GAD in DSM V) for example? "In particular, sense of coherence, which is a way of viewing life as predictable, manageable, and meaningful, can lower the risk for poor health outcomes.[18]"- Why Sense of coherence? Why not resilience for example? Although I can agree with the choice, it would be interesting to introduce the topic through a more general approach on coping strategies and to explain the reasons to select SOC as coping strategy over and above the others.- Please cite Antonovsky's first study on SOC (Antonovsky, A. (1987). Unraveling the mystery of health. How people manage
---

stress and stay well. San Francisco, CA: Lossey-Bass Publishers.)

- A helpful reference on coping strategies could be Fossion et al. (Fossion, P., Leys, C., Kempnaers, C., Braun, S., Verbanck, P., & Linkowski, P. (2014). Disentangling Sense of Coherence and Resilience in case of multiple traumas. *Journal of Affective Disorders*, 160, 21-26. DOI : 10.1016/j.jad.2014.02.029)

“The objective of this study will be to determine whether SOC moderates the link between area deprivation and generalized anxiety disorder in women using a large, longitudinal, population cohort”

- If authors choose a moderation analysis, does it mean that they consider that SOC is a stable personality trait rather than a dynamic aptitude? Is deprivation considered as a trauma? If yes is it a single or a multiple trauma? If it is a single trauma is it a type I or II (see Terr’s classification)? In my opinion, a moderation is adequate if deprivation is seen as a type II trauma (single but chronic trauma) that will not erode SOC ability (or trait?) to a sensitization mechanism (see literature on the subject, such as Snekkvik, H., Anke, A. G., Stanghelle, J. K., & Fugl-Meyer, A. R. (2003). Is sense of coherence stable after multiple trauma?. *Clinical Rehabilitation*, 17(4), 443-453. Or Fossion, P., Leys, C., Kempnaers, C., Braun, S., Verbanck, P., & Linkowski, P. (2013). Depression, anxiety and loss of resilience after multiple traumas: an illustration of a mediated moderation model of sensitization in a group of children who survived the Nazi Holocaust. *Journal of affective disorders*, 151(3), 973-979). However, if you consider deprivation as a multiple trauma and SOC as a skill, then, the traumas can erode SOC abilities and the possibility of a mediation rather than a moderation should be considered.
- Several studies emphasize the link between trauma, or stress, and GAD and include SOC either as mediator or as moderator, what is the input of this study?

“based on the 1991 Census”

- Please provide a reference for the Census. How is deprivation measured?
- Deprivation might not be the sole trauma, but authors did not measure the presence/absence of other trauma in participants’ life. This issue could be addressed in limitation section.
- Please provide reference for HLEQ (Surtees, P. G., Wainwright, N. W. J. & Brayne, C. (2000) Psychosocial aetiology of chronic disease: a pragmatic approach to the assessment of lifetime affective morbidity in an EPIC component study. *Journal of Epidemiology and Community Health*, 54, 114– 122)
- Please specify to what extent HLEQ allows a valid diagnosis of GAD. I see that authors provide a list of criteria to diagnose GAD, but they do not provide any information on this process (is it a standard list of criteria? Please provide references)

“Covariates were chosen a priori based on previous literature”

- Please provide references for each covariate.
- Please explain why covariates such as the use of anti-depressors, anxiolytics, and such were not taken into account. I would suspect an effect of anxiolytics on GAD.

“The HLEQ included a three-item SOC questionnaire”

- Three items to measure three factors of a construct is a very

	poor way to assess a construct. If there is any ambiguity in the question (which is usually the case) you will fail to capture the concept. There is a validated questionnaire on 13 items usually used to assess SOC why was it not used? This is a serious limitation of the study. Besides please provide Cronbach alphas.  - The median split is also problematic; participants close to the median do not differ much between themselves. I see no reason not to keep this variable continuous. - Please discuss the odd ratios, since authors use z-scores, odd-ratios can be related to effect size and effect size is an important information if you want to understand the extent of the link between your variables. You need to consider that you have a very large sample size (which is a strength) and thus much statistical power. Therefore, a significant test is not very informative and it is important to rely on the effect sizes. “When the interaction between SOC and GAD was assessed, the p-value was 0.221.”  - It is unclear whether authors tested the interaction between deprivation and SOC (which should be done if you want to test a moderation). - I do not understand why nor how authors computed the interaction between SOC (the moderator) and GAD (the dependent variable)??? - It is important to clarify, because, in discussion, authors describe simple effects (i.e. the difference in the link between deprivation and GAD among women with low level of SOC compare to high level of SOC), however, if the interaction is not significant, these two simple effects are not different from each other even if one is significant and the other not. “Coping skills or sense of coherence (SOC) appeared to mediate the association between area deprivation and anxiety”  - In discussion, this sentence state that SOC is a mediator, whereas authors consider SOC as a moderator (even in the title). This makes no sense (see comments above). “Finally, the measure of SOC that we used in this study has been reported to be valid, reliable, and theoretically-sound [30].”  - I am sorry to put it bluntly but if I understood well the way authors measured SOC, this assertion is just not true. In the study of Walsh, McCartney, McCullough, Buchanan, and Jones cited by authors, SOC was measured through the usual 13 items scale. In this case it is measured with a three items scale that has not been validated, and that is very likely to have poor psychometric properties. I hope these comments will help authors to improve their paper, even if I am afraid that some shortcomings may be methodological and impossible to overcome. Best regards
--	--

REVIEWER	Samet Kose, MD, PhD University of Texas Medical School at Houston, Houston, TX, USA
REVIEW RETURNED	27-Oct-2017

GENERAL COMMENTS	Very informative and nicely written primary data manuscript documenting that sense of coherence (SOC) moderates the association between area deprivation and generalize anxiety in women. The study was carefully designed and conducted with an impressive sample size. Psychometric scales including Health and Life Experiences Questionnaire (HLEQ)'s Coherence Items and Medical Outcomes Study 36-Item Short Form (SF-36) are appropriate. Statistical analysis is well-planned and sufficient. There are some typos (e.g., derived on line 461 which should be deprived), which can be handled during proof correction stage. I'd recommend use of "gender" rather than "sex" in the manuscript. Conclusion: Accept
---

VERSION 1 – AUTHOR RESPONSE

Reviewer: 1

Reviewer Name: Christophe Leys

Institution and Country: Université Libre de Bruxelles - Belgium Competing Interests: None declared

Authors conducted a study on a cohort of 20921 participants that completed a questionnaire assessing generalised anxiety disorder (GAD), sense of coherence (SOC), deprivation and various potential covariates. They describe a moderation model showing that, in a deprivation environment, women with a low sense of coherence suffer from generalised anxiety disorders more often than women with a high sense of coherence (even when the effect of some potential confounding variables was neutralized).

Although the study bears on a very large sample size, many serious shortcomings jeopardize authors results. The main concerns being a (very) poor measure of SOC and a lack of innovation (the link between adversity and GAD is well known and the role of SOC has often been identified, although sometimes as mediator and sometimes as moderator). For these reasons I suggested rejection, the rest could be overcome by a major revision. Please find below some comments:

1. Please provide a short description of general anxiety disorders (definition, symptoms) and some statistics about its prevalence.

* Response:

This paper focuses solely on generalized anxiety disorder (GAD); therefore, we began by describing this disorder specifically. We commented on its main features and symptoms: "It is characterized by excessive and pervasive worry about a number of areas of life, and associated symptoms, such as, restlessness, irritability, muscle tension, sleep difficulties, and concentration problems."

We did the same in another paper on generalized anxiety disorder (GAD) that we had published in BMJ Open (Remes et al. BMJ Open 2017).

We feel it is inappropriate to begin the paper by describing anxiety disorders in general (because 'overall anxiety' or 'anxiety disorders in general' is not our focus and the paper loses its conciseness – we are focusing specifically on generalized anxiety disorder). This is also the approach we had adopted in a previous BMJ Open paper (Remes et al. BMJ Open 2017), and it is also the approach used by the BMJ itself.

Within the paper, we calculate the prevalence of GAD in the UK. As such, we feel it is inappropriate to discuss the prevalence of GAD in other parts of the world in the introduction, because we are focusing specifically on Britain.

2. Please explain why deprivation is relevant to study? Is it a trauma? Is it a significant source of stress? Is it something else? Why would it yield GAD? Why not, I don't know, depression, addiction, eating disorders or PTSD (that has been removed from GAD in DSM V) for example?

“In particular, sense of coherence, which is a way of viewing life as predictable, manageable, and meaningful, can lower the risk for poor health outcomes.[18]“

* Response:

Deprivation is relevant to study, because the residential environment or the places in which we live can impact our mental health (this was mentioned in the manuscript). We now elaborated on the results we obtained in an antecedent study (Remes et al. BMJ Open 2017) – in a previous paper, we showed that living in deprivation significantly increases the risk of GAD in women, but not in men. Some of the reasons accounting for this sex differential are women's greater exposure to their residential environment compared to men. Women tend to spend more time in their local communities, because they are more likely to have part-time work and carry out domestic or child-rearing duties. If women are living in an area with low socio-economic circumstances, they are thus more likely to be exposed to the stress and strain that arises from deprivation. Exposure to stress can then increase the risk for central nervous system dysfunction and hypothalamic-pituitary-adrenal axis dysregulation, which can lead to the development of GAD.

While we did not have room to elaborate on all these details because of word count restrictions, we now provide a concise justification as to why area deprivation can increase the risk for GAD within the paper. We also cite our previous paper (Remes et al. BMJ Open 2017) that has shown this. This then provides a segway into our research on coping mechanisms among women living in deprivation.

This is the newly-inserted text in the Introduction section:

“Previously, we showed that women living in deprivation have a significantly higher risk of generalized anxiety disorder, while this was not observed in men. If women are living in an area with low socio-economic circumstances, they are more likely to be exposed to the stress and strain that arises from deprivation. Exposure to stress can then increase the risk for central nervous system dysfunction and hypothalamic-pituitary-adrenal axis dysregulation, which can lead to the development of GAD.”

3. Why Sense of coherence? Why not resilience for example? Although I can agree with the choice, it would be interesting to introduce the topic through a more general approach on coping strategies and to explain the reasons to select SOC as coping strategy over and above the others.

- A helpful reference on coping strategies could be Fossion et al. (Fossion, P., Leys, C., Kempnaers, C., Braun, S., Verbanck, P., & Linkowski, P. (2014). Disentangling Sense of Coherence and Resilience in case of multiple traumas. *Journal of Affective Disorders*, 160, 21-26. DOI : 10.1016/j.jad.2014.02.029)

“The objective of this study will be to determine whether SOC moderates the link between area deprivation and generalized anxiety disorder in women using a large, longitudinal, population cohort”

* Response:

We examined sense of coherence, because other papers examining exposure to adverse circumstances (in our case, deprivation) and mental health had done the same. For example, a paper examining exposure to early childhood deprivation and later posttraumatic stress used SOC as a measure (van der Hal-van Raalte EAM et al. *Journal of Clinical Psychology* 2008).

Since there are a number of coping approaches in the literature (hardiness, resilience, optimism, mastery), introducing these before discussing SOC would make the paper lose its focus. This is not a paper discussing coping in general, and the merits of using SOC above others. There is also no room for such discussion, because of word count restrictions.

The reviewer indicated that we could cite one of his papers (Fossion et al.) on coping. Since the reviewer's paper is not a literature review, we are unable to cite it. The introduction of the latter study includes a brief description of SOC and resilience and touches upon their similarities/differences based on other sources (other studies are cited to support the arguments the authors are making). This paper could only be cited if we were using the results from the primary data analysis conducted by the authors. If we wanted to cite further literature on coping, systematic reviews on this topic and/or papers written by the researchers who developed those concepts would need to be referenced. Also, the introduction of the paper indicated by the reviewer only discusses SOC and resilience, but there are many other coping resources (ex. hardiness, optimism, etc.). It is beyond the scope of this manuscript to go into such detail – this is because we are not conducting a literature review on the topic of coping.

Thus, as per one of your previous comments, we are citing the work done by Aaron Antonovsky, the developer of the SOC concept and measure – because this is the construct that we used in our study. Discussing other coping strategies would take away from the focus of our paper. We are also citing other papers, which specifically used the SOC measure in their analyses (ex. van der Hal-van Raalte 2008 and other papers from the EPIC-Norfolk cohort which used the SOC measure).

4. Please cite Antonovsky's first study on SOC (Antonovsky, A. (1987). *Unraveling the mystery of health. How people manage stress and stay well.* San Francisco, CA: Jossey-Bass Publishers.)

* Response:

Done

5. If authors choose a moderation analysis, does it mean that they consider that SOC is a stable personality trait rather than a dynamic aptitude? Is deprivation considered as a trauma? If yes is it a single or a multiple trauma? If it is a single trauma is it a type I or II (see Terr's classification)? In my opinion, a moderation is adequate if deprivation is seen as a type II trauma (single but chronic trauma) that will not erode SOC ability (or trait?) to a sensitization mechanism (see literature on the subject, such as Snekkvik, H., Anke, A. G., Stanghelle, J. K., & Fugl-Meyer, A. R. (2003). Is sense of coherence stable after multiple trauma?. *Clinical Rehabilitation*, 17(4), 443-453. Or Fossion, P., Leys, C., Kempnaers, C., Braun, S., Verbanck, P., & Linkowski, P. (2013). Depression, anxiety and loss of resilience after multiple traumas: an illustration of a mediated moderation model of sensitization in a group of children who survived the Nazi Holocaust. *Journal of affective disorders*, 151(3), 973-979). However, if you consider deprivation as a multiple trauma and SOC as a skill, then, the traumas can erode SOC abilities and the possibility of a mediation rather than a moderation should be considered.

* Response:

The reviewer asked whether we consider SOC as a "stable personality trait" vs. a "dynamic aptitude". SOC has been defined in the literature as "a flexible and adaptive dispositional orientation enabling successful coping with adverse experience" (Antonovsky 1987). It represents one's capacity to deal with stressful situations and indicates how well one can adapt after being exposed to adversity. We have clarified this further in the Introduction.

Area deprivation is different from a trauma. Our measure of area deprivation was based on levels of unemployment, percentage of people without housing, percentage of people without a car, and

percentage of people living in overcrowded homes. Area deprivation is more akin to poverty – it refers to communities without basic necessities to carry on day-to-day living.

The reviewer, on the other hand, has based his comment on one of his previously published papers, which focused on multiple traumas and SOC as a “stable personality trait/dynamic aptitude”. In his study, the reviewer assessed exposure to traumas in a sample of 65 people—25 women and 40 men—who survived the Holocaust. He defines traumas as being exposed to: “1) forced separation from family and friends 2) poor caretaking, 3) impairment in development, and 4) a return to a hostile and anti-Semitic environment after WWII”.

The reviewer’s work is very different from our own and the demands/measurement issues that arise in epidemiology. The reviewer recruited a highly-select sample of people who were exposed to severe, extreme circumstances, such as forced separation from family and poor care-taking, which he defined as traumas (very different from our research examining community disadvantage). Area deprivation refers to the residential environment/surrounding community to which one is continuously exposed, and it is different from experiencing deeply distressing, discrete events (reviewer’s research). In our study, area deprivation refers to levels of unemployment, non-home ownership, non-car ownership, and overcrowding, which we indicated.

6. Several studies emphasize the link between trauma, or stress, and GAD and include SOC either as mediator or as moderator, what is the input of this study?

* Response:

This is the first study to assess whether coping mechanisms (ie. one’s level of sense of coherence) can buffer the harmful effect of living in a deprived area on mental health. Specifically, whether sense of coherence can mitigate the risk of generalized anxiety disorder among women living in deprivation. We have reviewed the literature and no prior study has examined this.

Other studies have indeed looked at the link between trauma/stress and anxiety; however, no prior study has assessed the moderating effect of SOC with respect to area deprivation and GAD. This is important to examine, because GAD is one of the most common mental health problems in society, and large numbers of people are living in deprivation around the world.

7. “based on the 1991 Census”

- Please provide a reference for the Census. How is deprivation measured?

* Response:

A reference for the Census has been provided. We specified how area deprivation is measured in the paper under the heading “Assessment of area deprivation-exposure”.

8. Deprivation might not be the sole trauma, but authors did not measure the presence/absence of other trauma in participants’ life. This issue could be addressed in limitation section.

* Response:

As mentioned in a previous comment, area deprivation is not a trauma. The aim of our paper was not to determine the links between previous traumas, sense of coherence, and GAD. We aimed to examine whether sense of coherence is an effect modifier of the association between area deprivation and anxiety. This manuscript builds on a previous paper published in BMJ Open (Remes et al. BMJ Open 2017), which examined the association between area deprivation and GAD. In that paper, there was no discussion of traumas, because this was not the focus of the research.

9. Please provide reference for HLEQ (Surtees, P. G., Wainwright, N. W. J. & Brayne, C. (2000) Psychosocial aetiology of chronic disease: a pragmatic approach to the assessment of lifetime affective morbidity in an EPIC component study. *Journal of Epidemiology and Community Health*, 54, 114– 122)

* Response:

Thank you for this comment. We have now provided a reference for the HLEQ.

10. Please specify to what extent HLEQ allows a valid diagnosis of GAD. I see that authors provide a list of criteria to diagnose GAD, but they do not provide any information on this process (is it a standard list of criteria? Please provide references)

* Response:

We specified in the text that GAD was constructed according to the DSM-IV (a standard process for measuring anxiety disorders). A structured self-assessment questionnaire (HLEQ) was used to capture symptoms of GAD. Also, we provided a reference for this (Surtees P, Wainwright N, Khaw K, Day N. Functional health status, chronic medical conditions and disorders of mood. *Br J Psychiatry* 2003;183:299–303).

11. "Covariates were chosen a priori based on previous literature"

- Please provide references for each covariate.

* Response:

Thank you for this comment. In the Methods section, we stated that covariates were chosen based on their association with anxiety and deprivation, and we have now provided the relevant references for this.

12. Please explain why covariates such as the use of anti-depressors, anxiolytics, and such were not taken into account. I would suspect an effect of anxiolytics on GAD.

* Response:

In the antecedent paper to this manuscript, in which we assessed the link between area deprivation and GAD, we did not include anxiolytics and anti-depressants as covariates. Also, none of the reviewers for that paper suggested such an inclusion. Furthermore, when we submitted another paper to *BMJ Open* (in press) on the links between GAD and health service use, we had included psychotropic medication. However, one of the reviewers recommended the removal of psychotropics. As such, we removed medication for mental disorders from this paper, as well. Moreover, the studies on coping/SOC and anxiety do not include such a covariate in their analyses.

However, in line with the reviewer's comment, we ran an additional analysis adjusting for the effect of psychotropics (antihypnotics). The effect estimates remained essentially unchanged (the effect estimates for women living in deprivation with a weak SOC and strong SOC, respectively: OR=2.00, 95% CI: 1.36, 2.93 and OR=1.28, 95% CI: 1.28 (0.76, 2.16)). Because we do not feel this supplementary analysis adds to our results, we have omitted it from the paper – we were also limited by word count restrictions.

13. "The HLEQ included a three-item SOC questionnaire"

- Three items to measure three factors of a construct is a very poor way to assess a construct. If there is any ambiguity in the question (which is usually the case) you will fail to capture the concept.

There is a validated questionnaire on 13 items usually used to assess SOC why was it not used? This is a serious limitation of the study. Besides please provide Cronbach alphas.

- The median split is also problematic; participants close to the median do not differ much between themselves. I see no reason not to keep this variable continuous.

* Response:

Please see our response to comment 19 below. We elaborate on why we used the 3-item SOC.

In addition, we dichotomized the SOC variable rather than keeping it continuous, as per previous literature (ex. Surtees, Wainwright, *Journal of Psychosomatic Research* 2006; Wainwright, Surtees *Journal of Epidemiology and Community Health* 2007; Langham, Russell, et al. *Journal of Gambling Studies* 2017) and to ensure sufficient cell size. Furthermore, previous research indicated that people scoring 1 or 2 on the SOC scale were deemed to have a 'strong SOC' (Langham, Russell, et al. *Journal of Gambling Studies* 2017) – in our paper, we did the same: those scoring 1 or 2 were identified as having strong SOC.

Finally, in the paper written by the developers of the short SOC scale (Lundberg 1995), it is indicated that the SOC can be analysed either as continuous or dichotomous; we dichotomized it in line with the considerations mentioned above.

As per the reviewer comment, we have now provided Cronbach's alpha and elaborated on this in the paper. We have expanded on this in the Discussion section to read the following: "Internal consistency of the three-items SOC scale, as measured by Chronbach's alpha, was 0.35. While the internal consistency of the shorter 3-item measure was low in this sample, this is likely to be partially due to the small number of scale items. Also, the original developers of the scale reported satisfactory short-term test-retest reliability and validity for the 3-item measure (Surtees 2006, Lundberg 1995)."

14. Please discuss the odd ratios, since authors use z-scores, odd-ratios can be related to effect size and effect size is an important information if you want to understand the extent of the link between your variables. You need to consider that you have a very large sample size (which is a strength) and thus much statistical power. Therefore, a significant test is not very informative and it is important to rely on the effect sizes.

* Response:

We apologize, but this comment is unclear.

15. "When the interaction between SOC and GAD was assessed, the p-value was 0.221."

- It is unclear whether authors tested the interaction between deprivation and SOC (which should be done if you want to test a moderation).

* Response:

Thank you for this comment. We made a typo, and should have written "the interaction between deprivation and SOC" (which was what we had analysed). We have now corrected this.

16. I do not understand why nor how authors computed the interaction between SOC (the moderator) and GAD (the dependent variable)???

* Response:

Please see comment above.

17. It is important to clarify, because, in discussion, authors describe simple effects (i.e. the difference in the link between deprivation and GAD among women with low level of SOC compare to high level of SOC), however, if the interaction is not significant, these two simple effects are not different from each other even if one is significant and the other not.

* Response:

The reviewer indicated in one of his earlier comments that “a significance test is not very informative and it is important to rely on the effect size” – in line with this comment, even if the p-value of the interaction between deprivation and SOC is not statistically significant, it is important to examine the effect sizes.

The effect estimates show that women with a strong sense of coherence appear to have low levels of anxiety while women with a weak sense of coherence tend to have high anxiety levels. These results shed light on the importance of SOC when it comes to mitigating the risks of anxiety, and represent an important starting point in such research. Thus, future studies should replicate our research with a larger number of anxiety cases (perhaps measuring ‘total’ or ‘any’ anxiety rather than individual DSM-IV disorders, such as GAD). We have now clarified and included a section on this in the Discussion.

We thank the reviewer for his comment; however, we would like to mention that interpretations of studies are not based solely on statistical tests of significance and p-values, but also on effect estimates. We provide both types of measures in this study.

18. “Coping skills or sense of coherence (SOC) appeared to mediate the association between area deprivation and anxiety”

- In discussion, this sentence state that SOC is a mediator, whereas authors consider SOC as a moderator (even in the title). This makes no sense (see comments above).

* Response:

Thank you for this comment. “Mediator” was a typo – we have now changed it to “moderate”.

19. “Finally, the measure of SOC that we used in this study has been reported to be valid, reliable, and theoretically-sound [30].”

- I am sorry to put it bluntly but if I understood well the way authors measured SOC, this assertion is just not true. In the study of Walsh, McCartney, McCullough, Buchanan, and Jones cited by authors, SOC was measured through the usual 13 items scale. In this case it is measured with a three items scale that has not been validated, and that is very likely to have poor psychometric properties.

* Response:

Satisfactory short-term test-retest reliability and validity have been reported for the three-item SOC measure, which we used in our study (Wainwright, Surtees, Welch, Luben, Khaw, Bungham 2007). A number of papers have been published using this specific measure, for example Agardh, Ahlbom, et al. International Journal of Epidemiology 2007; Super, Verschuren, et al. Journal of Epidemiology and Community Health 2013; Langham, Russell, et al. Journal of Gambling Studies 2017; Nilsen, Andel, et al. European Journal of Public Health, 2016; Herens, Bakker, et al. PLOS One 2016; Eriksson, Hilding, et al. Epidemiology/Health Services Research 2013.

The “abbreviated SOC-3 scale was developed by Lundberg and Peck (1995) for situations where it was not practical to use either the full SOC-29 or SOC-13. The scale has satisfactory reliability ($\alpha=0.61$) and validity (Lundberg and Peck 1995) and has been used in large health studies (Agardh

et al. 2003; Avlund et al. 2003 ; Bayard-Burfield et al. 2001 ; Jahnsen et al. 2002 ; van Loon et al. 2001)" (Langham, Russell et al. Journal of Gambling Studies 2017).

As previous research indicates, a scale with more items would have been too lengthy in a multipurpose survey, where questions focusing on other social and health variables are competing for space and time; this was the case in our study, EPIC-Norfolk. As such, a shorter version was created (Lundberg, Peck. European Journal of Public Health, 1995) capturing each of the 3 dimensions included in the SOC, namely manageability, meaningfulness, and comprehensibility.

When the EPIC-Norfolk study was designed, the decision was made to recruit a large enough sample in order to allow detailed analyses to be undertaken on several exposures and endpoints/outcomes. Over 21,000 participants received a psychosocial questionnaire to complete, which measured a range of factors related to mood and emotions, including coping abilities. Because it was impractical and far too time-consuming to include a longer version (ex. the 13-item or the original 29-item version of the SOC instrument developed by Antonovsky) as part of a psychosocial questionnaire that was already extensive, the principal investigators included the three-item version. Several papers using the shorter SOC version have been published by EPIC-Norfolk (ex. Surtees, Wainwright, et al. American Journal of Epidemiology, 2003; Surtees, Wainwright, et al. Health Psychology, 2006; Surtees, Wainwright, Khaw Journal of Psychosomatic Research 2006) and others – please see first paragraph.

The three-item version covers the theoretical essence of each of the dimensions developed by Antonovsky (Lundberg, Peck. European Journal of Public Health, 1995).

Reviewer: 2

Reviewer Name: Samet Kose, MD, PhD

Institution and Country: University of Texas Medical School at Houston, Houston, TX, USA

Competing Interests: None declared

Very informative and nicely written primary data manuscript documenting that sense of coherence (SOC) moderates the association between area deprivation and generalize anxiety in women. The study was carefully designed and conducted with an impressive sample size. Psychometric scales including Health and Life Experiences Questionnaire (HLEQ)'s Coherence Items and Medical Outcomes Study 36-Item Short Form (SF-36) are appropriate. Statistical analysis is well-planned and sufficient.

There are some typos (e.g., derived on line 461 which should be deprived), which can be handled during proof correction stage. I'd recommend use of "gender" rather than "sex" in the manuscript.

* Response:

Thank you very much for this positive feedback. We have made the changes, as suggested.

VERSION 2 – REVIEW

REVIEWER	christophe leys université libre de bruxelles
REVIEW RETURNED	19-Dec-2017
GENERAL COMMENTS	Authors did not comply with comment 1 but provided reasons for not

doing so. Although to my opinion it is important to show that a research has important stakes (for example by commenting the prevalence of the DV at the beginning of the introduction) I believe this issue is up to the editor's appreciation. Concerning general anxiety disorder definition I agree that it is about generalized anxiety disorder, and that was what I ment (my mistake) in the comment, so authors did well to provide the definition about the right DV.

Authors did respond adequately with comment 2.

Authors did not comply with comment 3 but provided reasons for not doing so. Although I can agree with the hurdle of word count restriction, I still believe it is important to provide rationales explaining why a specific theoretical frame is chosen among many others. However, Authors specify SOC and I can agree that, given the word count, it is an acceptable compromise.

Authors did respond adequately with comment 4.

Authors did not comply with comment 5 but provided reasons for not doing so. Although they argue that our work are not related to this current work, this is not the point : they still do not justify convincingly why they used a moderation rather than a mediation model where as I believe it is important because it questions the whole process of SOC.

Authors provide arguments to justify the input of their study (comment 6) by stating the specificity of deprivation and the stakes related. I can agree with their argument now that they have specify this stakes in introduction (in response of comment 2).

Authors did respond adequately with comment 7.

Authors did not comply with comment 8 but provided reasons for not doing so. I can agree with authors justifications.

Authors did respond adequately with comment 9.

Authors did respond adequately to comment 10.

Authors did respond adequately to comment 11.

Authors did respond adequately to comment 12.

Authors did respond to comment 13 but I still believe this is the main shortcoming of the study. Authors base their assessment of SOC (a 3 factor concept) on a 3-item measure, yielding a Cronbach alpha of .35 (even with 3 items this is a very unsatisfying reliability). Moreover, they dichotomize the continuous scale which is often a (well known) problem (see for example : Cohen, J. (1983). The cost of dichotomization. *Applied Psychological Measurement*, 7(3), 249–253 ; Fitzsimons, G. J. (2008). Editorial: Death to dichotomizing. *Journal of Consumer Research*, 35(1), 5–8 ; McClelland, G. H., Lynch, J. G., Jr., Irwin, J. R., Spiller, S. A., & Fitzsimons, G. J. (2015). Median splits, type II errors, and false positive consumer psychology: Don't fight the power. *Journal of Consumer Psychology*. ; Rucker, D. D., McShane, B. B., & Preacher, K. J. (2015). A researcher's guide to regression, discretization, and median splits of continuous variables. *Journal of Consumer Psychology*.)

In my opinion, arguing that all this is fine because other authors did

	the same does not temper the issue. I really do not trust the measure, I am not at all convinced that it captures the SOC and I have nothing in my hands to know what it captures. So at the very least I expect some discussion around that matter in the text. Authors did not respond to comment 14 because they did not understand what I ment. I will try to rephrase : statistical significance is only one side of the coin in statistical decision. Having a very large sample implies that even small differences will be significant, i. e. not due to randomness. However, the other side of the coin is the effect size. A small effect is not as interesting as a medium or large effect. This effect size is not reported in author's analysis but they refer to odd ratios that can be used as an estimation of the effect size. Now in the text, the difference between high level of SOC (OR = 1.42) and low level of SOC (OR = 1.93) is a very small difference (even if significant given the large sample size). So I believe it is interesting to address this in discussion : there is an effect but a small effect (just so that people do not conclude that it would be wise to invest much in therapies that bolster SOC to overcome the effect of deprivation on GAD for example). Authors did respond adequately to comment 15.` Authors did respond adequately to comment 16. Authors did respond to comment 17. However, although I agree that p-value and effect size are both equally important (see above), if interaction is not significant (and again, with such a large sample, even very small effect sizes would be detected), I find it very hasardous to consider the simple effects. Yet, authors addressed the issue in discussion, not exactly in the way I would have done it but it is still fine by me. Authors did respond adequately to comment 18. Authors did respond to comment 19. However, as stated above I am still very doubtfull about the measure. At least authors report a validation for their 3-item scale which is an improvement, although the reliability is very poor in this case and it is still a major concern. In sum, I believe authors reviewed their paper adequately, but the main shortcomings remain because they seem to be inherent to the design (and they have not been addressed in limitations). I will suggest rejection because it is easier for an editor to turn rejection into acceptation than to turn acceptation into rejection, but it is really up to the editor to assess whether the extend of the shortcomings is acceptable or not in regard with the standards of the review.
--	---

VERSION 2 – AUTHOR RESPONSE

Editorial Comments:

1. We appreciate that the reviewer's comments below are still quite negative; however, we felt we should give you the opportunity to respond to the remaining criticisms and revise your manuscript appropriately.

We urge you to respond to all comments as thoroughly as possible. Please address the concerns by clearly outlining the limitations of your study design and methods in both the 'Strengths and limitations' section and the 'Discussion' section.

Response:

Thank you very much for giving us the opportunity to respond to the reviewer comments.

As per the editorial comments, we addressed the concerns by including the following in the first paragraph of the Discussion:

“SOC was based on a simplified 3-item measure, with modest internal reliability (Cronbach's alpha=0.35)[22], and this variable was dichotomized. Although it may be useful to additionally employ a continuous SOC measure, we dichotomized this variable, because previous literature had done so as well.[24]”

Also, under the Strengths and Limitations section, we elaborate on this and repeat the shortcoming of this study. This is the excerpt:

“It should also be mentioned that the internal consistency of the three-items SOC scale, as measured by Cronbach's alpha, was 0.35.[22] While the internal consistency of the shorter 3-item measure was low in this sample, this is likely to be partially due to the small number of scale items. Also, the original developers of the scale reported satisfactory short-term test-retest reliability and validity for the 3-item measure.[22,41] Despite this, it was a limitation that we did not have a longer measure with higher reliability, such as the SOC-13 or SOC-29.[41] The SOC was dichotomized – it may be useful to additionally use a continuous measure of SOC, but we dichotomized it because research on coping had done so as well.[24] ”

In keeping with literature on coping, we included a binary/dichotomous variable of SOC. However, in line with reviewer 1's comments, we mention that it may have been useful to also include a continuous measure of this construct (although it would have been less meaningful, because we wanted to separate women with a weak and strong SOC).

Reviewer: 1

Reviewer Name: Christophe Leys

Institution and Country: Université libre de bruxelles, Belgium

Competing Interests: None declared

1. Authors did not comply with comment 1 but provided reasons for not doing so. Although to my opinion it is important to show that a research has important stakes (for example by commenting the prevalence of the DV at the beginning of the introduction) I believe this issue is up to the editor's appreciation. Concerning general anxiety disorder definition I agree that it is about generalized anxiety disorder, and that was what I ment (my mistake) in the comment, so authors did well to provide the definition about the right DV.

* Thank you.

2. Authors did respond adequately with comment 2.

* Thank you.

3. Authors did not comply with comment 3 but provided reasons for not doing so. Although I can agree with the hurdle of word count restriction, I still believe it is important to provide rationales explaining why a specific theoretical frame is chosen among many others. However, Authors specify SOC and I can agree that, given the word count, it is an acceptable compromise.

* Thank you.

4. Authors did respond adequately with comment 4.

* Thank you.

5. Authors did not comply with comment 5 but provided reasons for not doing so. Although they argue that our work are not related to this current work, this is not the point : they still do not justify convincingly why they used a moderation rather than a mediation model where as I believe it is important because it questions the whole process of SOC.

Authors provide arguments to justify the input of their study (comment 6) by stating the specificity of deprivation and the stakes related. I can agree with their argument now that they have specify this stakes in introduction (in response of comment 2).

* As per the reviewer comments, we tested a mediation model as well, and such a model does not hold in our analysis. Key steps in mediation analysis include the following: 1) a model of the mediator as a function of the predictor (M <-- X model), and 2) a model of the response as a function of both the mediator and the predictor (Y <-- MX model). If the effect of X in the first model and the effect of M in the second model are both significant, then there is evidence of mediation. Model 1: when we regressed area deprivation against SOC in a fully-adjusted model, the effect (of the explanatory variable: area deprivation) was not statistically significant (p-value=0.372). Model 2: when we included area deprivation and SOC as separate variables in a fully-adjusted model with GAD as the outcome,

the p-values were 0.0005 (for deprivation) and <0.0001 (for SOC). However, because the relationship between area deprivation and SOC was not statistically significant in Model 1, we could not go forward with a mediation analysis. We have now added in the following text in the Results section (in case readers should wonder whether mediation is possible):

“When area deprivation was regressed against SOC in a fully-adjusted model, the p-value was 0.372; and when area deprivation and SOC were introduced in a fully-adjusted model with GAD as the outcome, the p-values for these explanatory variables were 0.0005 and <0.0001, respectively.”

The second reason for undertaking a moderation analysis is a theoretical one. We were not interested in whether area deprivation influences SOC, and whether SOC subsequently increases risk of having GAD. We wanted to know whether having a strong SOC changes the strength of the relationship between area deprivation and anxiety – we wanted to assess the moderating effects of SOC.

We were not interested in the mechanism as to why area deprivation might lead to anxiety (this is a different question, and there could be several mechanisms), but rather: if women have strong coping skills, does area deprivation still lead to increased risk for GAD? Is the relationship between area deprivation and GAD different depending on how weak or strong women’s SOC is? This was what we were interested in, and it was a theoretical decision.

6. Authors did respond adequately with comment 7.

* Thank you.

7. Authors did not comply with comment 8 but provided reasons for not doing so. I can agree with authors justifications.

* Thank you.

8. Authors did respond adequately with comment 9.

*Thank you.

9. Authors did respond adequately to comment 10.

* Thank you.

10. Authors did respond adequately to comment 11.

* Thank you.

11. Authors did respond adequately to comment 12.

* Thank you.

12. Authors did respond to comment 13 but I still believe this is the main shortcoming of the study. Authors base their assessment of SOC (a 3 factor concept) on a 3-item measure, yielding a Cronbach alpha of .35 (even with 3 items this is a very unsatisfying reliability). More over, they dichotomize the continuous scale which is often a (well known) problem (see for example : Cohen, J. (1983). The cost of dichotomization. *Applied Psychological Measurement*, 7(3), 249–253 ; Fitzsimons, G. J. (2008). Editorial: Death to dichotomizing. *Journal of Consumer Research*, 35(1), 5–8 ; McClelland, G. H., Lynch, J. G., Jr., Irwin, J. R., Spiller, S. A., & Fitzsimons, G. J. (2015). Median splits, type II errors, and false positive consumer psychology: Don't fight the power. *Journal of Consumer Psychology*. ; Rucker, D. D., McShane, B. B., & Preacher, K. J. (2015). A researcher's guide to regression, discretization, and median splits of continuous variables. *Journal of Consumer Psychology*.)

In my opinion, arguing that all this is fine because other authors did the same does not temper the issue. I really do not trust the measure, I am not at all convinced that it captures the SOC and I have nothing in my hands to know what it captures. So at the very least I expect some discussion around that matter in the text.

* Thank you for this comment. We have now added in to the Discussion section (under Strengths and Limitations), that “it was a limitation that we did not have a longer measure with higher reliability, such as the SOC-13 or SOC-29.[41] The SOC was dichotomized – it may be useful to additionally use a continuous measure of SOC, but we dichotomized it because research on coping had done so as well.[24]”

In keeping with the reviewer's comment, we indicate that it may be useful to also include a continuous SOC measure. Also, it is appropriate to categorize variables in line with the literature. This was what we had also done in the paper antecedent to this: we categorized variables as per the literature (Remes O, *BMJ Open* 2017) - as such, we feel it is appropriate to dichotomize the SOC in keeping with previous research (ex. Langham, Russell, et al. *Journal of Gambling Studies* 2017). We re-iterate the limitations pointed out by the reviewer in the first paragraph of the Discussion as well, where we state: “SOC was based on a simplified 3-item measure, with modest internal reliability (Cronbach's alpha=0.35)[22], and this variable was dichotomized. Although it may be useful to additionally employ a continuous SOC measure, we dichotomized this variable because previous literature had done so as well.[24]”

It was a theoretical rather than analytical decision to dichotomize the SOC. Just as we conducted analyses between area deprivation and GAD for women and men separately (Remes O, *BMJ Open* 2017) because of theoretical reasons, we were interested in determining whether area deprivation increases risk of having GAD for women with a weak SOC and then separately for those with a strong SOC. Previous research indicated that people scoring 1 or 2 on the SOC scale were deemed to have a 'strong SOC' (Langham, Russell, et al. *Journal of Gambling Studies* 2017) – in our paper, we did the same: those scoring 1 or 2 were identified as having strong SOC.

We would like to conclude by mentioning one additional comment regarding the utility of the simplified SOC measure we used. In the paper by Lundberg and Peck (1995), the developers of the 3-item SOC, they mention the following:

"The simplified SOC indicator appears to be reasonably valid in relation to the theoretical propositions given by Antonovsky and highly valid in that it relates to health and other variables in an expected way." For further details regarding the research conducted on this, please see Lundberg and Peck, European Journal of Public Health 1995.

13. Authors did not respond to comment 14 because they did not understand what I meant. I will try to rephrase: statistical significance is only one side of the coin in statistical decision. Having a very large sample implies that even small differences will be significant, i. e. not due to randomness. However, the other side of the coin is the effect size. A small effect is not as interesting as a medium or large effect. This effect size is not reported in author's analysis but they refer to odd ratios that can be used as an estimation of the effect size. Now in the text, the difference between high level of SOC (OR = 1.42) and low level of SOC (OR = 1.93) is a very small difference (even if significant given the large sample size). So I believe it is interesting to address this in discussion: there is an effect but a small effect (just so that people do not conclude that it would be wise to invest much in therapies that bolster SOC to overcome the effect of deprivation on GAD for example).

* Thank you for this comment. We have updated our Discussion, accordingly. We now indicate that the differences between women with low and high SOC are rather small, as seen below:

"Although the interaction between area deprivation and SOC was not statistically significant, the effect estimates do suggest that there are differences between women with low and high SOC – nevertheless, these differences are rather small."

The effect estimates changed slightly after running correlated data analysis (previously we did not take the intra-class correlation into account); however, we do mention that the differences between women with weak and strong SOC are rather small, as suggested.

14. Authors did respond adequately to comment 15.

* Thank you.

15. Authors did respond adequately to comment 16.

* Thank you.

16. Authors did respond to comment 17. However, although I agree that p-value and effect size are both equally important (see above), if interaction is not significant (and again, with such a large sample, even very small effect sizes would be detected), I find it very hazardous to consider the

simple effects. Yet, authors addressed the issue in discussion, not exactly in the way I would have done it but it is still fine by me.

* Thank you.

17. Authors did respond adequately to comment 18.

* Thank you.

18. Authors did respond to comment 19. However, as stated above I am still very doubtful about the measure. At least authors report a validation for their 3-item scale which is an improvement, although the reliability is very poor in this case and it is still a major concern.

* We have now added in the Discussion that, "it was a limitation that we did not have a longer measure with higher reliability, such as the SOC-13 or SOC-29."

In sum, I believe authors reviewed their paper adequately, but the main shortcomings remain because they seem to be inherent to the design (and they have not been addressed in limitations). I will suggest rejection because it is easier for an editor to turn rejection into acceptance than to turn acceptance into rejection, but it is really up to the editor to assess whether the extent of the shortcomings is acceptable or not in regard with the standards of the review.